# The Antimicrobial Peptides Human β-Defensins Induce the Secretion of Angiogenin in Human Dermal Fibroblasts

**DOI:** 10.3390/ijms23158800

**Published:** 2022-08-08

**Authors:** Yoshie Umehara, Miho Takahashi, Hainan Yue, Juan Valentin Trujillo-Paez, Ge Peng, Hai Le Thanh Nguyen, Ko Okumura, Hideoki Ogawa, François Niyonsaba

**Affiliations:** 1Atopy (Allergy) Research Center, Juntendo University Graduate School of Medicine, Tokyo 113-8421, Japan; 2Department of Dermatology and Allergology, Juntendo University Graduate School of Medicine, Tokyo 113-8421, Japan; 3Faculty of International Liberal Arts, Juntendo University, Tokyo 113-8421, Japan

**Keywords:** human β-defensin (hBD), dermal fibroblast, angiogenin, angiogenesis

## Abstract

The skin produces a plethora of antimicrobial peptides that not only show antimicrobial activities against pathogens but also exhibit various immunomodulatory functions. Human β-defensins (hBDs) are the most well-characterized skin-derived antimicrobial peptides and contribute to diverse biological processes, including cytokine production and the migration, proliferation, and differentiation of host cells. Additionally, hBD-3 was recently reported to promote wound healing and angiogenesis, by inducing the expression of various angiogenic factors and the migration and proliferation of fibroblasts. Angiogenin is one of the most potent angiogenic factors; however, the effects of hBDs on angiogenin production in fibroblasts remain unclear. Here, we investigated the effects of hBDs on the secretion of angiogenin by human dermal fibroblasts. Both in vitro and ex vivo studies demonstrated that hBD-1, hBD-2, hBD-3, and hBD-4 dose-dependently increased angiogenin production by fibroblasts. hBD-mediated angiogenin secretion involved the epidermal growth factor receptor (EGFR), Src family kinase, c-Jun N-terminal kinase (JNK), p38, and nuclear factor-kappa B (NF-κB) pathways, as evidenced by the inhibitory effects of specific inhibitors for these pathways. Indeed, we confirmed that hBDs induced the activation of the EGFR, Src, JNK, p38, and NF-κB pathways. This study identified a novel role of hBDs in angiogenesis, through the production of angiogenin, in addition to their antimicrobial activities and other immunomodulatory properties.

## 1. Introduction

The skin is the primary interface between the body and the surrounding environment. To defend against pathogens, skin produces hundreds of antimicrobial peptides that exhibit antimicrobial activity against bacteria, fungi, and viruses. In addition to their antimicrobial activity, antimicrobial peptides regulate inflammatory responses, cytokine/chemokine secretion, and cell migration and proliferation, and improve skin barrier function [1,2]. Two major groups of antimicrobial peptides, defensins and cathelicidins, have been well characterized in human skin [3]. Based on sequence homology and the connectivity of conserved cysteine (Cys) residues, human defensins are classified into α-defensins and β-defensins. The characteristic connection of disulfide bridges in human α-defensins is Cys^1^–Cys^6^, Cys^2^–Cys^4^, and Cys^3^–Cys^5^, while that in human β-defensins (hBDs) is Cys^1^–Cys^5^, Cys^2^–Cys^4^, and Cys^3^–Cys^6^ [4]. Human α-defensin-1, -2, -3, and -4 are also termed human neutrophil peptides (HNPs), because they are mainly expressed in neutrophils. HNP-1, -2, and -3 differ only in the first amino acid, whereas HNP-4 has a distinct amino acid sequence [5,6]. hBDs are some of the most important skin-derived antimicrobial peptides and are well known for their wide range of microbicidal activities and immunomodulatory properties [7]. Six hBDs have been identified, among which hBD-1, hBD-2, hBD-3, and hBD-4 are primarily found in the epithelium of the skin, eyes, and oral, respiratory, and urogenital tracts, while hBD-5 and hBD-6 are exclusively expressed in the epididymis [7]. In normal skin, hBD-1 is constitutively expressed, whereas the expression of hBD-2, hBD-3, and hBD-4 is induced in lesional skin by injury, infection, or inflammation [7]. As hBDs activate various types of host cells, such as keratinocytes, fibroblasts, mast cells, neutrophils, and macrophages, hBDs greatly contribute to biological processes. It has been reported that hBDs are involved in pro- and anti-inflammatory responses, neutralization of lipopolysaccharides, chemoattraction, activation of autophagy, maintenance of the skin barrier, and promotion of wound healing [1,3,7,8,9]. In cutaneous wound healing, which occurs in healthy subjects, hBD-2 and hBD-3 are especially expressed by keratinocytes at wound sites [8]. hBD-3 was recently reported to promote wound healing by modulating inflammatory responses, angiogenesis, and cell proliferation and migration in dermal fibroblasts [10]. Angiogenesis, which is the formation of new blood vessels, is orchestrated by angiogenic factors and is an important process in development, as well as wound healing [11]. A recent study showed that hBD-3 induced the production of angiogenic factors, including vascular endothelial growth factor (VEGF), platelet-derived growth factors (PDGF), and fibroblast growth factor (FGF), by dermal fibroblasts [10]. Angiogenin (ANG), a member of the RNase family that was initially discovered to be a tumor angiogenesis factor, is one of the most potent angiogenic factors and promotes the growth, survival, migration, and invasion of endothelial cells [12].

Although antimicrobial peptide derived from insulin-like growth factor-binding protein 5 (AMP-IBP5) was recently reported to increase ANG production by human epidermal keratinocytes [13], there have been no reports on hBD-induced secretion of ANG. Therefore, this study investigated the effects of the antimicrobial peptides hBD-1, hBD-2, hBD-3, and hBD-4 on the production of ANG in human dermal fibroblasts, with both in vitro and ex vivo models. The secretion of angiogenin was dose-dependently increased by hBD-1, hBD-2, hBD-3, and hBD-4, and this secretion was mediated by activation of the epidermal growth factor receptor (EGFR), Src family kinase, c-Jun N-terminal kinase (JNK), p38, and nuclear factor-kappa B (NF-κB) signaling pathways. These findings identified a novel role of hBDs in angiogenesis, through the production of ANG in dermal fibroblasts.

## 2. Results

### 2.1. hBDs Induce ANG Production by Human Dermal Fibroblasts in Both In Vitro and Ex Vivo Models

Normal human dermal fibroblasts were incubated with 10–20 μg/mL hBD-1, hBD-2, hBD-3, and hBD-4, and both the mRNA expression and extracellular secretion of ANG at different time points were examined. The mRNA expression of *ANG* and the amounts of ANG secreted in the cell-free culture supernatants were evaluated by quantitative real-time polymerase chain reaction (PCR) and enzyme-linked immunosorbent assay (ELISA), respectively. We observed that incubation with hBDs had little effect on the mRNA expression of *ANG* (Appendix A), whereas all hBDs dose- and time-dependently enhanced the secretion of ANG by fibroblasts (Figure 1a). A five-fold increase in ANG levels was observed at 6 h poststimulation, whereas a 10-fold increase was observed at 12 h. hBD-1, hBD-3, and hBD-4 significantly induced the production of ANG as early as 3 h, and hBD-3 displayed the strongest effect at concentrations as low as 10 μg/mL (Figure 1a).

Fibroblasts in vivo reside in a three-dimensional context. The three-dimensional matrix culture allows fibroblasts to grow into tissues that closely resemble their in vivo counterparts [14]. To determine whether hBDs induce ANG production under physiological conditions, normal human dermal fibroblasts were cultured in an ex vivo model for two days using type I collagen gels as scaffolds [15,16,17]. Fibroblasts were then stimulated with 20 μg/mL hBD-1, hBD-2, hBD-3, and hBD-4, and the amounts of ANG in the cell-free culture supernatants were evaluated using ELISA. The secretion of ANG by ex vivo-cultured fibroblasts was significantly increased following the addition of hBD-1, hBD-2, hBD-3, and hBD-4 (Figure 1b). The hBD-1-mediated stimulatory effect was only observed at 6 h poststimulation, whereas stimulation with hBD-2, hBD-3, and hBD-4 resulted in a time-dependent secretion of large amounts of ANG as early as 3 h after stimulation (Figure 1b). These findings suggested that hBDs are potent inducers of ANG secretion by human dermal fibroblasts, both in vitro and ex vivo.

The observation that all hBDs displayed an ANG-inducing effect prompted us to examine whether the ANG production was limited to hBDs. Therefore, we treated human fibroblasts with human α-defensins, whose disulfide linkages are different from those in hBDs. Among the human α-defensins tested, only HNP-4 induced ANG production. The effect of HNP-4 was dose- and time-dependent and was observed as early as 3 h after stimulation (Figure 2a). Other α-defensins, such as HNP-1, -2, and -3, did not show any effect (data not shown). In addition to defensins, we examined the ANG-inducing effect of cathelicidin LL-37, which does not contain disulfide linkages in its sequence. We found that LL-37 also significantly induced ANG secretion. Therefore, it appears that hBD-mediated ANG production is not dependent on the presence of disulfide bridges in hBDs, and that among antimicrobial peptides, ANG production is not limited to hBDs.

### 2.2. EGFR and Src Family Kinase Activation Is Necessary for the hBD-Mediated Production of ANG in Dermal Fibroblasts

The EGFR signaling pathway regulates fundamental cellular functions, including cell survival, proliferation, and migration [18]. Since hBDs have been shown to activate hu man keratinocytes via EGFR [19], we examined the role of EGFR signaling in the hBD-mediated ANG production by human dermal fibroblasts. First, we confirmed that hBDs activate EGFR. As shown in Figure 3a, the phosphorylation of EGFR increased between 30 and 120 min after stimulation with hBD-1, hBD-2, hBD-3, and hBD-4. To examine whether increased EGFR activation was indeed necessary for hBD-mediated ANG production, fibroblasts were preincubated with an EGFR-specific inhibitor, AG1478, for 2 h before stimulation with each hBD. As shown in Figure 3a, the ANG secretion induced by hBDs was suppressed by pretreatment with AG1478, confirming that hBDs enhanced ANG production via the EGFR pathway.

It has been reported that the Src family tyrosine kinases are activated by tyrosine kinase receptors and promote signaling through various growth factor receptors, including EGFR [20,21,22]. As hBDs promoted the activation of EGFR, we examined the effects of hBDs on the Src signaling pathway in fibroblasts. Figure 4a shows that all hBDs induced Src phosphorylation. This activation was first observed at 30 min poststimulation and peaked at 120 min. Moreover, pretreating fibroblasts with the Src family tyrosine kinase inhibitor PP2 significantly suppressed hBD-induced ANG production (Figure 4b), indicating the involvement of the Src signaling pathway in hBD-mediated ANG secretion. The observation that treatment of fibroblasts with either EGFR inhibitor (AG1478) or Src inhibitor (PP2) only partially inhibited hBD-induced ANG production suggests that pathways other than EGFR and Src may be involved in hBD-induced secretion of ANG by human fibroblasts. We confirmed that both AG1478 (Appendix A) and PP2 (Appendix A) completely inhibited EGFR and Src phosphorylation, respectively, indicating that the incubation time and doses of inhibitors used in this study were sufficient to inhibit the EGFR and Src pathways.

### 2.3. hBD-Mediated ANG Production Requires the Activation of the Mitogen-Activated Protein Kinase (MAPK) and Nuclear Factor-Kappa B (NF-κB) Pathways

The MAPK pathways play key roles in many biological activities, including cell survival and metabolism, and have been reported to be activated by hBDs in keratinocytes and mast cells [23,24,25]. To evaluate whether hBDs could also activate MAPKs in fibroblasts, cells were incubated with hBD-1, hBD-2, hBD-3, and hBD-4 for 5–60 min, and the phosphorylation of MAPK p38, JNK, and extracellular signal-regulated kinase (ERK) 1/2 was examined using Western blotting. hBD-1, hBD-2, hBD-3, and hBD-4 increased the phosphorylation of p38 at 5 min, and this phosphorylation was still remarkable at 60 min after stimulation with hBD-1 and hBD-3. All hBDs also enhanced JNK and ERK1/2 phosphorylation, which peaked at 5 min, before decreasing (Figure 5a). The requirement for MAPK pathways in hBD-mediated ANG production was evaluated by treating fibroblasts with MAPK inhibitors for 2 h, before stimulation with hBDs. SB203580 (p38 inhibitor) and JNK inhibitor II markedly decreased hBD-mediated ANG production, while U0126 (ERK inhibitor) had no effect on ANG secretion (Figure 5b). The failure of U0126 to inhibit hBD-induced ANG production by fibroblasts was not due to the inactivity of this inhibitor, because treatment of fibroblasts with U0126 completely suppressed both hBD-induced and spontaneous ERK phosphorylation. Other inhibitors, SB203580 and JNK inhibitor II, also abolished p38 and JNK phosphorylation (Appendix A).

In addition, we focused on the NF-κB pathway, because this pathway has been implicated in antimicrobial peptide-mediated cell activation [26,27], and hBDs have been demonstrated to activate the NF-κB signaling pathway in human epidermal keratinocytes and cervical cancer cells [28,29]. To determine whether hBDs activated NF-κB signaling in fibroblasts and thereby promoted the nuclear translocation of NF-κB, NF-κB expression in the nucleus was analyzed after hBD stimulation of fibroblasts. hBD-1, hBD-2, hBD-3, and hBD-4 significantly increased the levels of NF-κB in the nucleus at 60 min poststimulation (Figure 6a). The observation that hBD-mediated ANG secretion was suppressed by pretreatment with NF-κB activation inhibitor II suggests that the NF-κB signaling pathway is necessary for hBD-mediated ANG secretion (Figure 6b). We confirmed that the nuclear translocation of NF-κB was suppressed in fibroblasts following treatment of cells with NF-κB activation inhibitor II (Appendix A). Collectively, these data demonstrated that the MAPK JNK and p38 and NF-κB signaling pathways are necessary for the hBD-mediated production of ANG by fibroblasts.

## 3. Discussion

A recent study indicated that the skin-derived antimicrobial peptide hBD-3 promoted wound healing, angiogenesis, proliferation, and migration in fibroblasts in an in vivo mouse model and in cultured primary human dermal fibroblasts. Furthermore, hBD-3 promoted angiogenesis by enhancing the dermal fibroblast production of angiogenic factors, such as VEGF, PDGF, and FGF [10]. In this study, we demonstrate that the secretion of another angiogenic factor ANG by fibroblasts was increased not only by hBD-hBD-1, hBD-2, hBD-3, and hBD-4. hBD-mediated ANG production involved the EGFR, Src, p38, JNK, and NF-κB signaling pathways.

As a first line of defense against microbial invasion, human skin produces antimicrobial peptides, which play crucial roles in both innate and adaptive immune responses during injury and inflammation. In addition to antimicrobial properties, hBDs exhibit diverse bioactivities and regulate cell proliferation and migration, skin barrier function, wound healing, and angiogenesis [7]. It has been reported that hBDs also induce the production of various cytokines, growth factors, and angiogenic factors and induce autophagy in epidermal keratinocytes [9,10,19,23,24]. Angiogenesis plays a critical role in many biological processes, such as development, tissue remodeling, reproduction, wound healing, and carcinogenesis. Although ANG is one of the most potent angiogenic factors, it has not been investigated whether hBDs regulate the production of ANG in human dermal fibroblasts. Here, we demonstrated that ANG production was greatly induced by hBD-1, hBD-2, hBD-3, and hBD-4 in fibroblasts cultured in both in vitro and ex vivo models.

As hBDs are induced in skin tissues following injury, infection, or inflammation, there may be an interaction between various resident skin cells, due to impairment of the skin structure and infiltration of inflammatory cells [8]. During the wound healing process, cellular interactions become dominated by the interplay between keratinocytes and fibro blasts, which gradually shift the microenvironment away from an inflammatory site toward synthesis-driven granulation tissue [30]. Of note, the effect of hBDs on fibroblasts has been investigated, to develop a potential agent for wound healing [31,32]. Moreover, the possibility that hBDs may be expressed in the dermis has been reported in lesional skin of acne vulgaris [33,34], and hBDs are upregulated in the dermis of chronic wounds [35], where they may directly interact with fibroblasts.

Although the exact physiological concentrations of hBDs are not well known, the doses of hBD-2 have been estimated to be 3.5–16 μM (15–70 μg/mL) in IL-1α-stimulated epidermal cultures [36] and approximately 20 μM (87 μg/mL) in skin tissues from patients with psoriasis [37]. Furthermore, high doses of hBDs have been detected in wound healing [38], where ANG plays a critical role [12]. Therefore, we assume that the doses of hBDs (20 μg/mL, equivalent to 4–5 μM) used in this study are physiologically relevant.

As all hBDs increased ANG secretion, one could speculate that hBD-mediated ANG production is dependent on hBD structure. In fact, the presence of disulfide bridges has been shown to be important in hBD-mediated immunomodulatory activities, although it is not indispensable for hBD-induced antimicrobial activities [39]. To investigate the importance of disulfide bridges in hBD-induced ANG secretion, fibroblasts were treated with α-defensins, whose disulfide linkages are different from those in hBDs. Only HNP-4, but not HNP-1, HNP-2, or HNP-3, induced ANG production. HNP-4 has also been previously reported to be more effective than other HNPs in protecting peripheral blood mononuclear cells from HIV-1 infection [40]. In addition, because LL-37, which is unable to form disulfide bonds, also induced ANG secretion by fibroblasts, it is assumed that hBD-mediated ANG production is independent of the presence of disulfide bridges in hBDs.

hBDs have been reported to activate the EGFR and MAPK pathways in human keratinocytes [19,23,28]. Accordingly, in this study, we observed that hBDs induced the phosphorylation of EGFR, Src and MAPK p38, JNK, and ERK1/2. Although the phosphorylation of ERK was strongly increased by hBDs, it seems that ERK activation was not involved in ANG production, because an ERK-specific inhibitor failed to suppress ANG production. In addition to EGFR, Src, and MAPK, hBD-induced ANG production was also mediated by the NF-κB signaling pathway. This finding is consistent with previous studies reporting that antimicrobial peptides activate the NF-κB signaling pathway [26,27,28,41]. However, the observation that EGFR and Src inhibitors partially suppress hBD-induced ANG suggests the possibility of other pathways with hBD-driven activities. In fact, hBDs have been reported to activate various physiological regulators of intracellular signaling pathways, such as reactive oxygen species, which are involved in ANG secretion [42,43]. Further studies are needed to clarify whether these pathways are involved in hBD-mediated ANG secretion.

The levels of angiogenic factors have been shown to be downregulated in nonhealing chronic wounds [44]. As a recent study indicated that hBD-3 induced the production of angiogenic factors, such as VEGF, PDGF, and FGF2, by dermal fibroblasts [10], and because this report demonstrated that hBDs promote ANG secretion by fibroblasts, these findings indicate that hBDs may contribute to angiogenesis, in addition to their antimicrobial and other immunomodulatory activities. Therefore, it is hypothesized that promoting the production or receptor activation of angiogenic factors may be a useful treatment strategy for chronic wounds. In conclusion, hBD treatment may be effective in promoting angiogenesis and wound healing through angiogenin secretion by fibroblasts.

## 4. Materials and Methods

### 4.1. Reagents

The antimicrobial peptides hBD-1, hBD-2, hBD-3, hBD-4, and HNP-4 were obtained from the Peptide Institute (Osaka, Japan) and dissolved in 0.01% acetic acid. LL-37 (L^1^LGDFFRKSKEKIGKEFKRIVQRIKDFLRNLVPRTES^37^) was synthesized by the solid-phase method, as previously reported [24]. AG1478 was purchased from Selleck (Houston, TX). PP2, JNK inhibitor II, U0126, SB203580, and NF-κB activation inhibitor II were obtained from Calbiochem (La Jolla, CA, USA). Can Get Signal^®^ Immunoreaction Enhancer Solution (TOYOBO Co., Ltd., Osaka, Japan) was used as the diluent for the antibodies.

The primary antibodies anti-EGFR (1:2000 dilution), anti-phospho-EGFR (1:1000 dilution), anti-SAPK/JNK (1:500 dilution), anti-phospho-SAPK/JNK (1:1000 dilution), anti-p38 (1:1000 dilution), anti-phospho-p38 (1:2000 dilution), anti-p44/42 (ERK1/2) (1:2000 dilution), anti-phospho-p44/42 (ERK1/2) (1:2000 dilution), and anti-NF-κB p65 (1:1000 dilution) were purchased from Cell Signaling Technology (Beverly, MA, USA), and anti-Lamin B1 (1:20,000 dilution) was obtained from Proteintech (Rosemont, IL, USA). Horseradish peroxidase (HRP)-linked anti-rabbit IgG and HRP-linked anti-mouse IgG secondary antibodies were purchased from Cytiva (Tokyo, Japan) and diluted at 1:2000.

### 4.2. Cell Culture and Stimulation

Primary human dermal fibroblasts isolated from neonatal foreskin were purchased from Lifeline Cell Technology (Osaka, Japan) and cultured in FibroLife Basal Medium (Lifeline Cell Technology) containing L-glutamine (7.5 mM), human FGF-basic (5 ng/mL), insulin (5 μg/mL), ascorbic acid (50 μg/mL), hydrocortisone (1 μg/mL), gentamycin (30 μg/mL), amphotericin B (15 ng/mL), and fetal bovine serum (2% vol/vol) at 37 °C with 5% CO_2_ and used within 3 passages. All experiments were performed using subconfluent cells grown in FibroLife Basal Medium without supplements, but with antibiotics, in a 12-well cell culture plate (Greiner Bio-One, Frickenhausen, Germany).

For the ex vivo model, primary human dermal fibroblasts were suspended in Dulbecco’s modified Eagle medium (DMEM, Sigma–Aldrich, St Louis, MO, USA) containing 0.15% atelocollagen (KOKEN, Tokyo, Japan) at a density of 5 × 10^5^ cells/mL, and 100 μL of cell suspension was seeded in a 96-well cell culture plate (Greiner Bio-One). After 1 h, DMEM supplemented with 10% fetal bovine serum (Biosera, Boussens, France) was added to the collagen gels. Two days later, the cells were used for studies to mimic ex vivo physiological conditions, as reported previously [14,15,16,17].

### 4.3. Preparation of Total RNA and Quantitative Real-Time PCR

Total RNA was extracted from cultured fibroblasts using a RNeasy Plus Micro Kit (Qiagen, Tokyo, Japan), according to the manufacturer’s guidelines. Samples were reverse-transcribed with ReverTra Ace qPCR RT Master Mix (TOYOBO, Osaka, Japan), according to the manufacturer’s protocol. Quantitative real-time PCR was performed on a StepOne Plus Real-time PCR System (Applied Biosystems, Foster City, CA, USA) using a QuantiNova SYBR Green PCR kit (Qiagen). Each sample was analyzed in duplicate; the amounts of ANG mRNA in each sample were normalized to those of ribosomal protein S18 (RPS18), and the expression of ANG mRNA was expressed relative to its expression in untreated control cells. The pairs of specific primers for ANG (forward primer: 5′-GTGCTGGGTCTGGGTCTGAC-3′; reverse primer: 5′-GGCCTTGATGCTGCGCTTG-3′), and RPS18 (forward primer: 5′-TTTGCGAGTACTCAACACCAACATC-3′; reverse primer: 5′-GAGCATATCTTCGGCCCACAC-3′).

### 4.4. ELISA

Fibroblasts were stimulated with 10–20 μg/mL hBDs for 3–24 h, and cell-free supernatants were collected. In some experiments, fibroblasts were pretreated for 2 h with 100 nM AG1478 (EGFR inhibitor), 20 μM PP2 (Src inhibitor), 10 μM SB203580 (p38 inhibitor), 10 μM JNK inhibitor II, 10 μM U0126 (ERK inhibitor), and 40 μM NF-κB activation inhibitor II, before stimulation with 20 μg/mL hBD-1, 20 μg/mL hBD-2, 10 μg/mL hBD-3, and 20 μg/mL hBD-4 for 6 h. In the preliminary dose-dependent experiments, the abovementioned concentrations were the most effective and less toxic (data not shown). ANG was quantified using DuoSet ELISA kits obtained from R&D Systems (Minneapolis, MN, USA).

### 4.5. Western Blotting

Following stimulation, fibroblasts were lysed with RIPA buffer (Cell Signaling Technology) containing phosphatase inhibitor cocktails 2 and 3 (Sigma-Aldrich, St. Louis, MO, USA). Nuclear lysates were prepared using a LysoPure™ Nuclear and Cytoplasmic Extractor Kit (FUJIFILM Wako Pure Chemical Corporation, Tokyo, Japan) containing protease inhibitor cocktail Set III (FUJIFILM Wako Pure Chemical Corporation). Protein levels were quantified using precision red advanced protein assay reagent #2 (Cytoskeleton, Denver, CO, USA). Equal amounts of total protein were separated on 10% SDS-polyacrylamide gels. After electrophoresis, the proteins were transferred onto an immobilon-P transfer membrane (Millipore, Billerica, MA, USA) by a PoweredBLOT 2M system (ATTO, Tokyo, Japan). The membranes were blocked with ImmunoBlock (KAC, Hyogo, Japan) for 1 h at room temperature and incubated with primary antibodies at 4 °C overnight. The membranes were washed with Tris-buffered saline with 0.1% Tween 20 and incubated with HRP-conjugated secondary antibodies. After being washed, the membranes were developed with Luminata Western HRP substrate (Millipore, Billerica, MA, USA), and the bands were detected with ImageQuant™ LAS 4000 (FUJIFILM Wako Pure Chemical Corporation). The intensity of the bands was quantified using ImageJ software (NIH, Bethesda, MD, USA).

### 4.6. Statistical Analysis

The data were analyzed using GraphPad Prism 8 (GraphPad Software Inc., San Diego, CA, USA). The differences between means were analyzed by one-way ANOVA with Tukey’s multiple comparison tests. In all analyses, *p* < 0.05 was considered statistically significant.

## Figures and Tables

**Figure 1 ijms-23-08800-f001:**
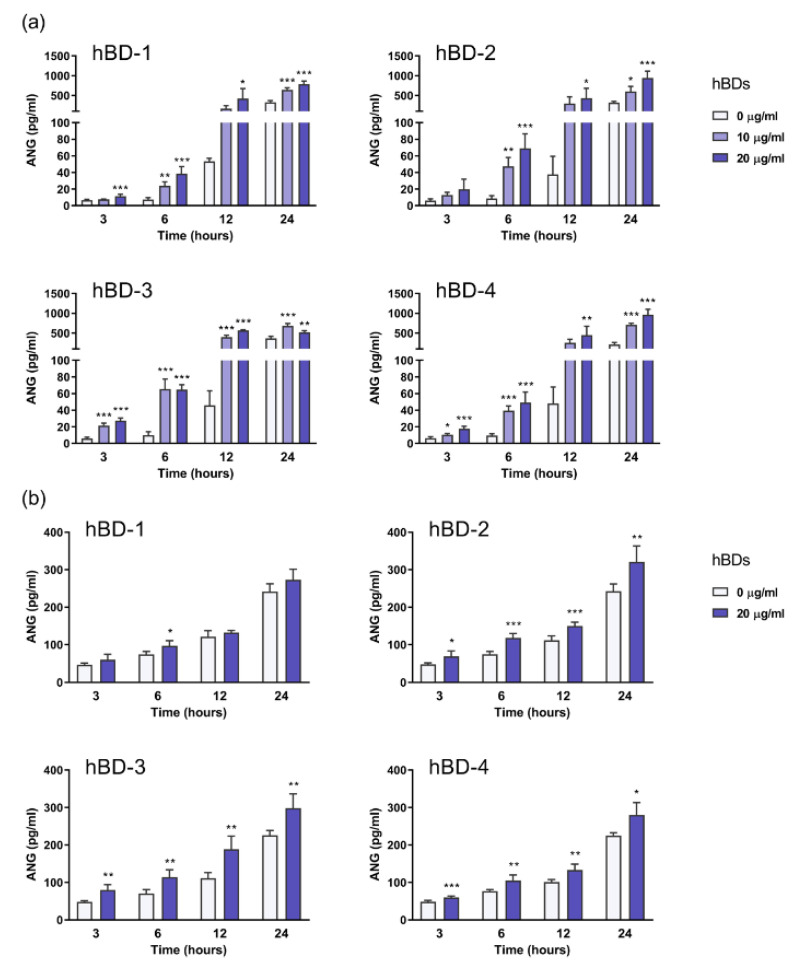
hBDs induce the secretion of the angiogenic factor ANG by human dermal fibroblasts. (**a**) Normal human dermal fibroblasts were stimulated with 10–20 μg/mL hBDs or vehicle (0 μg/mL hBDs) for 3–24 h, and the amounts of ANG in the culture supernatant were determined by ELISA. (**b**) Normal human dermal fibroblasts were cultured in a three-dimensional context using type I collagen gels and then stimulated with 20 μg/mL hBDs or vehicle (0 μg/mL hBDs) for 3–24 h. The amounts of ANG in the culture supernatant were determined by ELISA. The data represent the means ± SDs of 4–5 separate experiments. * *p* < 0.05, ** *p* < 0.01, and *** *p* < 0.001 compared with the vehicle at each time point by one-way ANOVA with Tukey’s multiple comparisons test.

**Figure 2 ijms-23-08800-f002:**
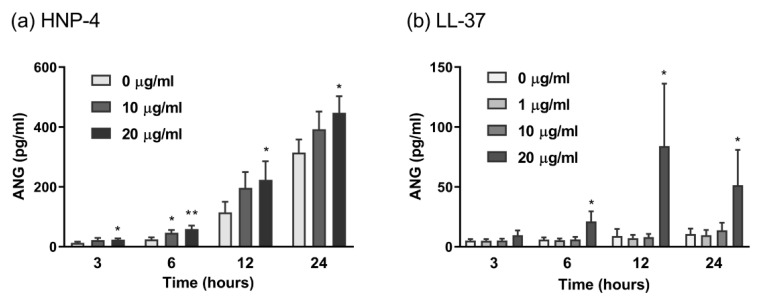
The production of ANG is promoted by HNP-4 and LL-37. Normal human dermal fibroblasts were cultured with 10–20 μg/mL HNP-4 (**a**) or 1–20 μg/mL LL-37 (**b**) for 3–24 h. The concentration of ANG in the culture supernatant was determined by ELISA. The data represent the means ± SDs of 3–4 separate experiments. * *p* < 0.05 and ** *p* < 0.01 compared with the vehicle at each time point by one-way ANOVA with Tukey’s multiple comparisons test.

**Figure 3 ijms-23-08800-f003:**
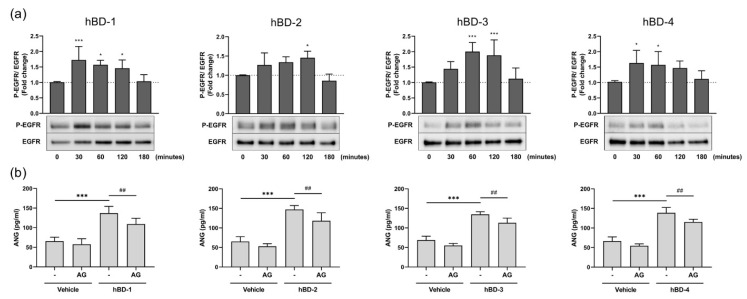
hBDs induce ANG production by activating the EGFR pathway. (**a**) Human dermal fibroblasts were stimulated with 20 μg/mL hBD-1, 20 μg/mL hBD-2, 10 μg/mL hBD-3, or 20 μg/mL hBD-4 for 30–120 min, and the levels of phosphorylated and unphosphorylated EGFR in whole cell lysates were analyzed by Western blotting. * *p* < 0.05 and *** *p* < 0.001 compared between the unstimulated (0 min) and the hBD-stimulated groups by one-way ANOVA with Tukey’s multiple comparisons test. (**b**) Fibroblasts were preincubated with 100 nM AG1478 (AG, EGFR inhibitor) or solvent (-) for 2 h. Cells were then stimulated for 6 h with 20 μg/mL hBD-1, 20 μg/mL hBD-2, 10 μg/mL hBD-3, or 20 μg/mL hBD-4. The concentrations of ANG in cell-free supernatants were determined by ELISA. *** *p* < 0.001 compared between the presence and absence of hBDs without inhibitors; ## *p* < 0.01 compared between the inhibitor-treated and untreated group with hBD stimulation by one-way ANOVA with Tukey’s multiple comparisons test. All results are the means ± SDs of 5–6 independent experiments.

**Figure 4 ijms-23-08800-f004:**
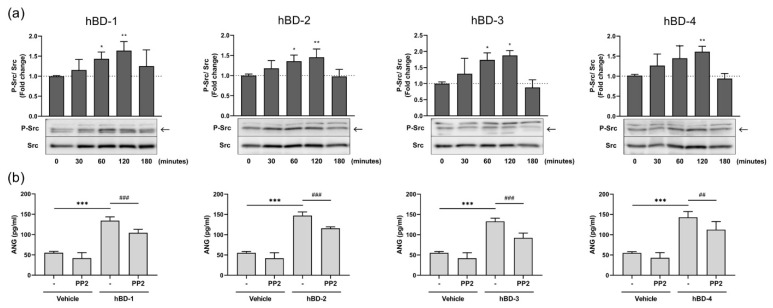
hBDs induce ANG production by activating the Src pathway. (**a**) Human dermal fibroblasts were stimulated with 20 μg/mL hBD-1, 20 μg/mL hBD-2, 10 μg/mL hBD-3, or 20 μg/mL hBD-4 for 30–120 min, and the levels of phosphorylated and unphosphorylated Src in whole cell lysates were analyzed by Western blotting. * *p* < 0.05 and ** *p* < 0.01 compared between the unstimulated (0 min) and the hBD-stimulated groups by one-way ANOVA with Tukey’s multiple comparisons test. (**b**) Fibroblasts were preincubated with 20 μM PP2 (Src inhibitor) or solvent (-) for 2 h. Cells were then stimulated for 6 h with 20 μg/mL hBD-1, 20 μg/mL hBD-2, 10 μg/mL hBD-3, or 20 μg/mL hBD-4. The concentrations of ANG in cell-free supernatants were determined by ELISA. *** *p* < 0.001 compared between the presence and absence of hBDs without inhibitors; ## *p* < 0.01 and ### *p* < 0.001 compared between the inhibitor-treated and untreated groups with hBD stimulation using one-way ANOVA with Tukey’s multiple comparisons test. All results are the means ± SDs of 3–6 independent experiments.

**Figure 5 ijms-23-08800-f005:**
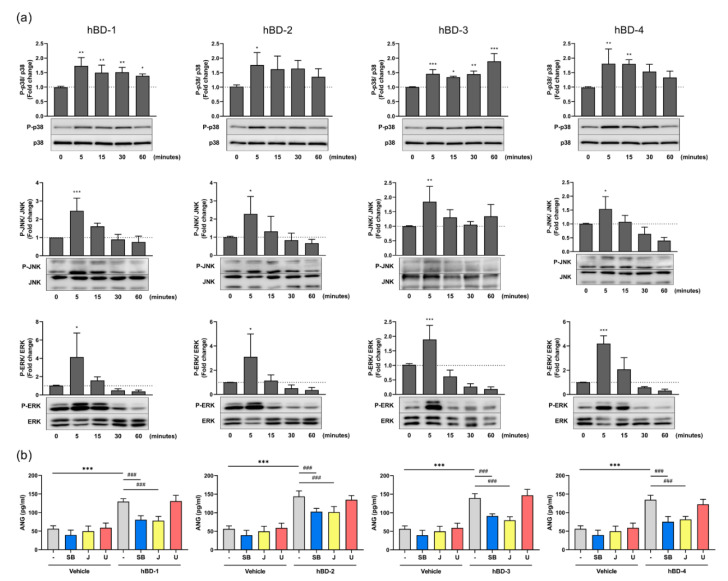
hBDs induce ANG production by activating the p38 and JNK signaling. (**a**) Human dermal fibroblasts were stimulated with 20 μg/mL hBD-1, 20 μg/mL hBD-2, 10 μg/mL hBD-3, or 20 μg/mL hBD-4 for 5–60 min, and the levels of phosphorylated and unphosphorylated MAPKs (p38, JNK and ERK1/2) in whole cell lysates were analyzed by Western blotting. * *p* < 0.05, ** *p* < 0.01 and *** *p* < 0.001 compared between the unstimulated (0 min) and the hBD-stimulated groups by one-way ANOVA with Tukey’s multiple comparisons test. (**b**) Fibroblasts were preincubated with 10 μM SB203580 (SB, p38 inhibitor), 10 μM JNK inhibitor II (J), 10 μM U0126 (U, ERK inhibitor), or solvent (-) for 2 h. Cells were then stimulated for 6 h with 20 μg/mL hBD-1, 20 μg/mL hBD-2, 10 μg/mL hBD-3, or 20 μg/mL hBD-4. The concentrations of ANG in cell-free supernatants were determined by ELISA. *** *p* < 0.001 compared between the presence and absence of hBDs without inhibitor; ### *p* < 0.001 compared between the inhibitor-treated and untreated group with hBD stimulation by one-way ANOVA with Tukey’s multiple comparisons test. All results are the means ± SDs of 3–6 independent experiments.

**Figure 6 ijms-23-08800-f006:**
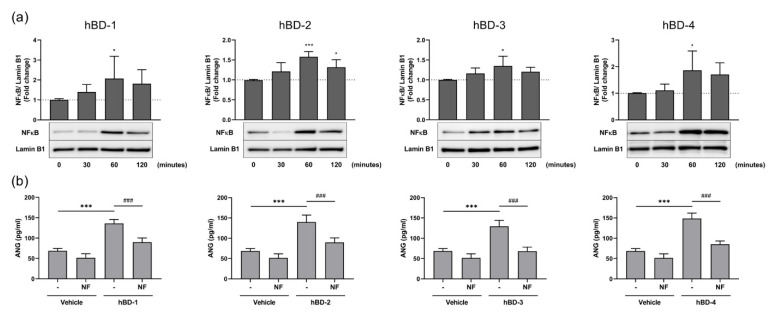
hBDs induce ANG production by activating NF-κB signaling. (**a**) Human dermal fibroblasts were stimulated with 20 μg/mL hBD-1, 20 μg/mL hBD-2, 10 μg/mL hBD-3, or 20 μg/mL hBD-4 for 5–60 min, and the levels of NF-κB in nuclear lysates were analyzed by Western blotting. The expression of Lamin B1 is shown as a loading control. * *p* < 0.05 and *** *p* < 0.001 compared between the unstimulated (0 min) and the hBD-stimulated groups by one-way ANOVA with Tukey’s multiple comparisons test. (**b**) Fibroblasts were preincubated with 40 μM NF-κB activation inhibitor II (NF) or solvent (-) for 2 h. Cells were then stimulated for 6 h with 20 μg/mL hBD-1, 10 μg/mL hBD-2, 10 μg/mL hBD-3, or 20 μg/mL hBD-4. The concentrations of ANG in cell-free supernatants were determined by ELISA. *** *p* < 0.001 compared between the presence and absence of hBDs without inhibitor; ### *p* < 0.001 compared between the inhibitor-treated and untreated groups with hBD stimulation by one-way ANOVA with Tukey’s multiple comparisons test. All results are the means ± SDs of 4–6 independent experiments.

## Data Availability

The data presented in this study are available upon request from the corresponding author.

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
