# Peer review of "The Antimicrobial Peptides Human β-Defensins Induce the Secretion of Angiogenin in Human Dermal Fibroblasts"

_ijms, 2022, doi:10.3390/ijms23158800_

Round 1
Reviewer 1 Report
The authors of the manuscript titled “The antimicrobial peptides human β-defensins induce the secretion of angiogenin in human dermal fibroblasts” present the work elucidating the role of human β-defensins in dermal fibroblasts. The authors have presented the significance of dose-dependent triggering of angiogenic pathways and interesting insights into the cross-talk with other related pathways. However, in spite of my initial interest for the work, there are many major concerns which diminish my enthusiasm for publication of the work.
- There should be a justification of the concentration of 10-20 µg/ml of hBDs used. Is this a physiologically relevant dosage?
- There have been very similar studies on keratinocytes. It is not clear where the novelty of this work lies?
- The experiments are repetitive and a lack of reflection on the true human physiology. The work should replicated in ex-vivo or in-vivo
- The results section in very sketchily written and there are only minimum supporting information and references.
- More information is necessary on how the experiments with inhibitors, AG1478 and SB203580 were performed. Was it a pretreatment? The timing and dosing should be elaborated.
- It is not clear why the defensins were directly administered to the fibroblasts. In most physiological scenario, there is a cross talk between keratinocytes and melanocytes, where secreted cytokines from keratinocytes affect the fibroblast activity. The possibility has been eliminated completely.
- The secretion of ANG is not enough evidence to support that this can lead to angiogenesis. There should have been other additional quantitation of bFGF, Pdgf, Vegfa expression or Immunofluorescence images in human skin to show these expressions. Without these information, it is premature to conclude “hBDs may contribute to angiogenesis in addition to their antimicrobial and immunomodulatory activities”-> Line 155,156.
Author Response
Reviewer 1
Comments and Suggestions for Authors
The authors of the manuscript titled “The antimicrobial peptides human β-defensins induce the secretion of angiogenin in human dermal fibroblasts” present the work elucidating the role of human β-defensins in dermal fibroblasts. The authors have presented the significance of dose-dependent triggering of angiogenic pathways and interesting insights into the cross-talk with other related pathways. However, in spite of my initial interest for the work, there are many major concerns which diminish my enthusiasm for publication of the work.
Answer
Dear Reviewer 1, we would like to thank you for taking the time to carefully read our manuscript and for your thoughtful comments that helped us to improve the manuscript. All of your comments have been addressed in the revised manuscript. In the following sections, please find a point-by-point reply to your comments. Our answers to your comments are in italics, and suggested changes to the text are highlighted in red.
Comment #1: There should be a justification of the concentration of 10-20 µg/ml of hBDs used. Is this a physiologically relevant dosage?
Answer to comment #1
Following your comment, we have discussed this point on Page 7, Lines 210–215 as follows: “Although the exact physiological concentrations of hBDs are not well-known, the concentrations of hBD-2 have been estimated to be 3.5–16 mM (15–70 mg/ml) in IL-1a-stimulated epidermal cultures [36] and approximately 20 mM (87 mg/ml) in skin tissues from patients with psoriasis [37]. Furthermore, high doses of hBDs have been detected in wound healing [38], where ANG plays a critical role [12]. Therefore, we assume that the doses of hBDs (20 mg/ml, equivalent to 4–5 μM) used in this study are physiologically relevant.”
Comment #2: There have been very similar studies on keratinocytes. It is not clear where the novelty of this work lies?
Answer to comment #2
The novelty of this work is described in this revised version as follows: “Although antimicrobial peptide derived from insulin-like growth factor-binding protein 5 (AMP-IBP5) was recently reported to increase ANG production by human epidermal keratinocytes [13], there are no reports on hBD-induced production of ANG” This is described on Page 2, Lines 69–71.
Comment #3: The experiments are repetitive and a lack of reflection on the true human physiology. The work should replicated in ex-vivo or in-vivo
Answer to comment #3
As recommended by the reviewer, ex vivo experiments were performed by culturing human fibroblasts in three-dimensional gels made of type I collagen. Following stimulation with hBDs, the secretion of ANG by fibroblasts was significantly increased. This is described in the Results section (Page 2, Lines 94–104) as follows: “Fibroblasts in vivo reside in a three-dimensional context. The three-dimensional matrix culture allows fibroblasts to grow into tissues that closely resemble their in vivo counterparts [14]. To determine whether hBDs induce ANG production under physiological conditions, normal human dermal fibroblasts were cultured in an ex vivo model for two days using type I collagen gels as scaffolds [15-17]. Fibroblasts were then stimulated with 20 μg/ml hBD-1, hBD-2, hBD-3 and hBD-4, and the amounts of ANG in the cell-free culture supernatants were evaluated by ELISA. The secretion of ANG by ex vivo-cultured fibroblasts was significantly increased following the addition of hBD-1, hBD-2, hBD-3 and hBD-4 (Figure 1b). hBD-1-mediated stimulatory effect was only observed at 6 hours poststimulation, whereas stimulation with hBD-2, hBD-3 and hBD-4 resulted in a time-dependent secretion of large amounts of ANG as early as 3 hours after stimulation (Figure 1b).” These findings suggested that hBDs are potent inducers of ANG secretion by human dermal fibroblasts both in vitro and ex vivo models.” This is also described in the Materials and Methods section (Page 9, Lines 277–283) as follows: “For the ex vivo model, primary human dermal fibroblasts were suspended in Dulbecco’s modified Eagle medium (DMEM, Sigma–Aldrich, St Louis, MO) containing 0.15% atelocollagen (KOKEN, Tokyo, Japan) at a density of 5×105 cells/ml, and 100 μl of cell suspension was seeded in a 96-well cell culture plate (Greiner Bio-One). After 1 hour, DMEM supplemented with 10% fetal bovine serum (Biosera, Boussens, France) was added to the collagen gels. Two days later, the cells were used for studies to mimic physiological conditions as reported previously [14-17].”
Comment #4: The results section in very sketchily written and there are only minimum supporting information and references.
Answer to comment #4
We deeply apologize for the lack of a detailed description of our results. In this revised version, we tried our best to ensure that all information essential for a good understanding of the results was provided.
Comment #5: More information is necessary on how the experiments with inhibitors, AG1478 and SB203580 were performed. Was it a pretreatment? The timing and dosing should be elaborated.
Answer to comment #5
As recommended by the reviewer, we provided necessary information regarding experiments with inhibitors. This is cited on Page 9, Lines 299–304 as follows: “In some experiments, fibroblasts were pretreated for 2 hours with 100 nM AG1478 (EGFR inhibitor), 20 μM PP2 (Src inhibitor), 10 mM SB203580 (p38 inhibitor), 10 mM JNK inhibitor II, 10 mM U0126 (ERK inhibitor) and 10 mM NF‐κB activation inhibitor II before stimulation with 20 μg/ml hBD-1, 20 μg/ml hBD-2, 10 μg/ml hBD-3 or 20 μg/ml hBD-4 for 6 hours. In the preliminary dose-dependent experiments, the abovementioned concentrations were more and less toxic (data not shown).”
Comment #6: It is not clear why the defensins were directly administered to the fibroblasts. In most physiological scenario, there is a cross talk between keratinocytes and melanocytes, where secreted cytokines from keratinocytes affect the fibroblast activity. The possibility has been eliminated completely.
Answer to comment #6
Thank you for your comment. In this study, defensins were directly administered to fibroblasts for the following reasons, as stated on Page, 6 Lines 200–209. “Because hBDs are induced in skin tissues following injury, infection or inflammation, there may be an interaction between various skin resident cells due to impairment of the skin structure and infiltration of inflammatory cells [8]. During the wound healing process, cellular interactions become dominated by the interplay between keratinocytes and fibroblasts, which gradually shift to the microenvironment away from an inflammatory site to a synthesis-driven granulation tissue [30]. Of note, the effect of hBDs on fibroblasts has been investigated to develop a potential agent for wound healing [31,32]. Moreover, the possibility that hBDs may be expressed in the dermis has been reported in lesional skin of acne vulgaris [33,34], and hBDs are upregulated in the dermis of chronic wounds [35], where they may directly interact with fibroblasts.
Comment #7: The secretion of ANG is not enough evidence to support that this can lead to angiogenesis. There should have been other additional quantitation of bFGF, Pdgf, Vegfa expression or Immunofluorescence images in human skin to show these expressions. Without these information, it is premature to conclude “hBDs may contribute to angiogenesis in addition to their antimicrobial and immunomodulatory activities”-> Line 155,156.
Answer to comment #7
Thank you for pointing out this issue. In this study, we did not investigate the effects of hBDs on bFGF, Pdgf, and Vegfa in human skin because in our recent report, we showed that hBDs induced the production of all abovementioned angiogenic factors by dermal fibroblasts [10]. In this revised version, we stated on Page 8, Lines 243–247 that “Because a recent study indicated that hBD-3 induced the production of angiogenic factors, such as VEGF, PDGF and FGF2, by dermal fibroblasts [10] and because this report demonstrated that hBDs also promote ANG secretion by fibroblasts, these findings indicate that hBDs may contribute to angiogenesis in addition to their antimicrobial and immunomodulatory activities.”
Reviewer 2 Report
Dear Editor, Dear Authors,
I was invited to evaluated the manuscript « The antimicrobial peptides human β-defensins induce the secretion of angiogenin in human dermal fibroblasts » by Yoshie Umehara et al.
In this study, the authors looked at the effect of human β-defensins (hBDs) that are produced by various tissues including skin contributing to diverse biological processes, including immunity against pathogens, immunomodulatory action, and wound healing effect. In relation to wound-healing effect of hBDs, the authors investigated their effects on the secretion of angiogenin using human dermal fibroblasts. Authors’s data show that hBDs caused a dose-dependent increase in secretion of angiogenin. Signal transduction assay using western-blot and the use of specific inhibitors further showed that hBDs activated various receptors and kinases including epidermal growth factor receptor (EGFR), Src family kinase, c-Jun N-terminal kinase (JNK), p38, extracellular signal-regulated kinase and nuclear factor-kappa B (NF-κB) pathways explaining potentially the effect on angiogenin secretion. Conclusions of the authors are that their study revealed the involvement of modulation of angiogenin in hBDs physiological effects other than antimicrobial or immunomodulatory activities.
Overall the study is very well conducted and the manuscript well written. I will have however few questions :
1- Did the authors looked at angiogenin gene expression (measurement of mRNA) and the effect of hBDs on it ? This is needed to see if hBDs effect is due to increase in both expression/secretion or only on secretion.
2- hBDs activities are most of the time dependent of the presence of disulfide bridges into those peptides. Did the authors explored the need of S-S bridges in angiogenin effect ? To do that, the authors would have to use modified hBD (using the one the more active) in which Cys residues are remplaced by ABU residues. This will be a bonus to the study.
3- Since all hBDs possess an effect on angiogenin secretion, it may also observed with other AMPs similar to hBDs. Did the authors looked at the effect of other BDs (not human) or of human alpha defensins or defensins from other animal species ?
Regards
Author Response
Reviewer 2
Comments and Suggestions for Authors
I was invited to evaluated the manuscript «The antimicrobial peptides human β-defensins induce the secretion of angiogenin in human dermal fibroblasts» by Yoshie Umehara et al.
In this study, the authors looked at the effect of human β-defensins (hBDs) that are produced by various tissues including skin contributing to diverse biological processes, including immunity against pathogens, immunomodulatory action, and wound healing effect. In relation to wound-healing effect of hBDs, the authors investigated their effects on the secretion of angiogenin using human dermal fibroblasts. Authors’s data show that hBDs caused a dose-dependent increase in secretion of angiogenin. Signal transduction assay using western-blot and the use of specific inhibitors further showed that hBDs activated various receptors and kinases including epidermal growth factor receptor (EGFR), Src family kinase, c-Jun N-terminal kinase (JNK), p38, extracellular signal-regulated kinase and nuclear factor-kappa B (NF-κB) pathways explaining potentially the effect on angiogenin secretion. Conclusions of the authors are that their study revealed the involvement of modulation of angiogenin in hBDs physiological effects other than antimicrobial or immunomodulatory activities.
Overall the study is very well conducted and the manuscript well written. I will have however few questions:
Answer
Dear Reviewer 2, we would like to thank you for taking the time to carefully read our manuscript and for your thoughtful comments that helped us to improve the manuscript. All of your comments have been addressed in the revised manuscript. In the following sections, please find a point-by-point reply to your comments. Our answers to your comments are in italics, and suggested changes to the text are highlighted in red.
Comment #1: Did the authors looked at angiogenin gene expression (measurement of mRNA) and the effect of hBDs on it ? This is needed to see if hBDs effect is due to increase in both expression/secretion or only on secretion.
Answer to comment #1
Thank you for pointing this out. In this revised version, we examined whether in addition to ANG secretion, hBDs may also induce gene expression. The result is described on Page 2, Lines 87–89 as follows: “We observed that incubation with hBDs had little effect on the mRNA expression of ANG (Supplementary Figure 1), whereas all hBDs dose- and time-dependently enhanced the secretion of ANG by fibroblasts (Figure 1a).”
Comment #2: hBDs activities are most of the time dependent of the presence of disulfide bridges into those peptides. Did the authors explored the need of S-S bridges in angiogenin effect? To do that, the authors would have to use modified hBD (using the one the more active) in which Cys residues are remplaced by ABU residues. This will be a bonus to the study.
Answer to comment #2
We completely agree with the reviewer that the presence of disulfide bridges is important in hBD-mediated immunomodulatory activities, although it has been shown that disulfide bonding is dispensable for hBD-3-induced antimicrobial properties [39]. The hBDs used in this study were purchased from the Peptide Institute. We have contacted the company to discuss the possibility of modifying these peptides by replacing Cys residues with ABU residues; however, the company refused our offer because the process would be costly and time-consuming. To investigate whether ANG production is limited to hBDs, we treated fibroblasts with human α-defensins (human neutrophil peptides, HNPs), whose disulfide linkages (Cys1-Cys6, Cys2-Cys4 and Cys3-Cys5) are different from those in hBDs (Cys1-Cys5, Cys2-Cys4 and Cys3-Cys6). Among human α-defensins, only α-defensin-4 (HNP-4) induced ANG production. In addition, because LL-37, which does not contain disulfide linkages, also induced ANG secretion, it appears that hBD-mediated ANG production is not dependent of the presence of disulfide bridges in hBDs. We described this issue on Page 7, Lines 216–227 as follows: “Because all hBDs increased ANG secretion, one can imagine that hBD-mediated ANG production is dependent of hBD structure. In fact, the presence of disulfide bridges has been shown to be important in hBD-mediated immunomodulatory activities, although it is not indispensable for hBD-induced antimicrobial activities [39]. To investigate the importance of disulfide bridges in hBD-induced ANG secretion, fibroblasts were treated with α-defensins, whose disulfide linkages are different from those in hBDs. Only HNP-4, but not HNP-1, HNP-2 or HNP-3, induced ANG production. HNP-4 has also been previously reported to be more effective than other HNPs in protecting peripheral blood mononuclear cells from HIV-1 infection [40]. In addition, because LL-37, which is unable to form disulfide bonds, also induced ANG secretion by fibroblasts, it appears that hBD-mediated ANG production is not dependent on the presence of disulfide bridges in hBDs.”
Comment #3: Since all hBDs possess an effect on angiogenin secretion, it may also observed with other AMPs similar to hBDs. Did the authors looked at the effect of other BDs (not human) or of human alpha defensins or defensins from other animal species?
Answer to comment #3
Following the reviewer’s recommendation, we examined the effects of human α-defensins and cathelicidin LL-37 on ANG production. Among the human α-defensins (HNPs) tested, only HNP-4 significantly induced ANG production, whereas HNP-1, -2 and -3 did not show any effect. Moreover, we observed that cathelicidin LL-37 also significantly increased ANG secretion by fibroblasts. This is stated on Page 7, Lines 216–227 as follows: “Because all hBDs increased ANG secretion, one can imagine that hBD-mediated ANG production is dependent of hBD structure. In fact, the presence of disulfide bridges has been shown to be important in hBD-mediated immunomodulatory activities, although it is not indispensable for hBD-induced antimicrobial activities [39]. To investigate the importance of disulfide bridges in hBD-induced ANG secretion, fibroblasts were treated with α-defensins, whose disulfide linkages are different from those in hBDs. Only HNP-4, but not HNP-1, HNP-2 or HNP-3, induced ANG production. HNP-4 has also been previously reported to be more effective than other HNPs in protecting peripheral blood mononuclear cells from HIV-1 infection [40]. In addition, because LL-37, which is unable to form disulfide bonds, also induced ANG secretion by fibroblasts, it appears that hBD-mediated ANG production is not dependent on the presence of disulfide bridges in hBDs.”
Reviewer 3 Report
Umehara et al, build on there previous work showing hBD3 promotes wound healing and angiogenesis through promoting the secretion of angiogenesis factors in dermal fibroblasts. Here they show hBDs induce the expression of angiogenin (ANG) in fibroblasts through MAPK/Src/NFKb pathways
The manuscript is well written but there are a few issues that need to be addressed
The effects on ANG production with the EGFR, Src, JNK and p38 inhibitors are small. The authors pre treat the cells for 2 hrs prior to a 6hr stimulation with the hBCs this is a short time frame for the inhibitors to work. The authors provide no evidence that the inhibitors work.
The authors would need to show pEGFR or pERK levels were reduced in the EGFR inhibitor treated cells in figure 2B, pSRC (tyr416) levels were reduced in the SRc inhibitor treated cells in figure 3b, pc-jun levels were reduced in the jnk inhibitor treated cells and p P38 levels were reduced in the p38 inhibitor cells ( both in figure 4) as well as showing the nfkb inhibitor worked in figure 5.
In the conclusion the authors state that ERK plays no role in hBC mediated ANG production but have not shown that U0126 has worked in the experimental condition. It could be the treatment time was not long enough for the inhibitor to work.
In fig 6 the authors should add that EGFR may activate JNK and p38 through MEK. In addition the authors provide no experimental evidence that JNK and p38 are downstream of GPCR.
Author Response
Reviewer 3
Comments and Suggestions for Authors
Umehara et al, build on there previous work showing hBD3 promotes wound healing and angiogenesis through promoting the secretion of angiogenesis factors in dermal fibroblasts. Here they show hBDs induce the expression of angiogenin (ANG) in fibroblasts through MAPK/Src/NFKb pathways
The manuscript is well written but there are a few issues that need to be addressed
Answer
Dear Reviewer 3, we would like to thank you for taking the time to carefully read our manuscript and for your thoughtful comments that helped us to improve the manuscript. All of your comments have been addressed in the revised manuscript. In the following sections, please find a point-by-point reply to your comments. Our answers to your comments are in italics, and suggested changes to the text are highlighted in red.
Comment #1: The effects on ANG production with the EGFR, Src, JNK and p38 inhibitors are small. The authors pre treat the cells for 2 hrs prior to a 6hr stimulation with the hBDs this is a short time frame for the inhibitors to work. The authors provide no evidence that the inhibitors work.
The authors would need to show pEGFR or pERK levels were reduced in the EGFR inhibitor treated cells in figure 2B, pSRC (tyr416) levels were reduced in the SRc inhibitor treated cells in figure 3b, pc-jun levels were reduced in the jnk inhibitor treated cells and p P38 levels were reduced in the p38 inhibitor cells ( both in figure 4) as well as showing the nfkb inhibitor worked in figure 5.
Answer to comment #1
Following the reviewer’s recommendations, we performed experiments to provide evidence of the inhibitors used in this study. As shown in Supplementary Figures 2–5, we confirmed that 100 nM AG1478 (EGFR inhibitor, Supplementary Figure 2), 20 mM PP2 (Src inhibitor, Supplementary Figure 3), 10 μM SB203580 (p38 inhibitor), 10 μM JNK inhibitor II and 10 μM U0126 (ERK inhibitor) (Supplementary Figure 4) and 40 mM NF-kB activation inhibitor II (Supplementary Figure 5) significantly inhibited the expression of their respective target proteins. This was cited on Page 4, Lines 139–145 as follows: “The observation that treatment of fibroblasts with either EGFR inhibitor (AG1478) or Src inhibitor (PP2) only partially inhibited hBD-induced ANG production suggests that pathways other than EGFR and Src may be involved in hBD-induced secretion of ANG by human fibroblasts. We confirmed that both AG1478 (Supplementary Figure 2) and PP2 (Supplementary Figure 3) completely inhibited EGFR and Src phosphorylation, respectively, indicating that the incubation time and doses of inhibitors used in this study are sufficient to inhibit the EGFR and Src pathways”.
Comment #2: In the conclusion the authors state that ERK plays no role in hBD mediated ANG production but have not shown that U0126 has worked in the experimental condition. It could be the treatment time was not long enough for the inhibitor to work.
Answer to comment #2
Following the reviewer’s comment, we performed experiments to confirm whether the doses and pretreatment time of U0126 were adequate to support our conclusion that ERK plays no role in hBD-mediated ANG production in fibroblasts. As shown in Supplementary Figure 4c, pretreatment of fibroblasts with 10 μM U0126 for 2 h before the addition of hBDs completely inhibited both spontaneous and hBD-induced ERK phosphorylation. This is stated in the revised version on Page 5, Lines 156–164 as follows: “The requirement of MAPK pathways in hBD-mediated ANG production was evaluated by treating fibroblasts with MAPK inhibitors for 2 hours before stimulation with hBDs. SB203580 (p38 inhibitor) and JNK inhibitor II markedly decreased hBD-mediated ANG production, while U0126 (ERK inhibitor) had no effect on ANG secretion (Figure 5b). The failure of U0126 to inhibit hBD-induced ANG production by fibroblasts was not due to inactivity of this inhibitor because treatment of fibroblasts with U0126 completely suppressed both hBD-induced and spontaneous ERK phosphorylation. Other inhibitors, SB203580 and JNK inhibitor II, also abolished p38 and JNK phosphorylation (Supplementary Figure 4).”
Comment #3: In fig 6 the authors should add that EGFR may activate JNK and p38 through MEK. In addition the authors provide no experimental evidence that JNK and p38 are downstream of GPCR.
Answer to comment #3
We completely agree with the reviewer that we should indicate that EGFR may activate JNK and p38 and that we did not provide experimental evidence that JNK and p38 are downstream of GPCR. We apologize for having provided a misleading figure. To avoid confusion, this figure was deleted from the revised version.
We corrected the discussion on Page 8, Lines 236–241 as follows: “However, the observation that EGFR and Src inhibitors partially suppress hBD-induced ANG suggests the possibility of other pathways with hBD-driven activities. In fact, hBDs have been reported to activate various physiological regulators of intracellular signaling pathways, such as reactive oxygen species, which are involved in ANG secretion [42,43]. Further studies are needed to clarify whether these pathways are involved in hBD-mediated ANG secretion”.
Round 2
Reviewer 1 Report
The authors have now addressed the queries sufficiently. The manuscript is fit for publication.
Reviewer 3 Report
The authors have address my concerns regarding the activity of the inhibitors used in this study. The authors have now clearly shown by western blot that the inhibitors are working in the assays described. The authors have addressed all my concerns.